# Security Quantification of Container-Technology-Driven E-Government Systems

**Subrota Kumar Mondal** [1,*,†] **, Tian Tan** [1,†]**, Sadia Khanam** [2]**, Keshav Kumar** [3]**, Hussain Mohammed Dipu Kabir** [4] **and Kan Ni** [1]

1 School of Computer Science and Engineering, Macau University of Science and Technology, Taipa, Macao 999078, China
2 Dhaka Dental College, University of Dhaka, Dhaka 1206, Bangladesh
3 University Institute of Computing, Chandigarh University, Mohali 140413, Punjab, India
4 Institute for Intelligent Systems Research and Innovation (IISRI), Deakin University, Geelong, VIC 3216, Australia
* Correspondence: skmondal@must.edu.mo
† These two authors contributed equally to this work.

**Abstract:** With the rapidly increasing demands of e-government systems in smart cities, a myriad of challenges and issues are required to be addressed. Among them, security is one of the prime concerns. To this end, we analyze different e-government systems and find that an e-government system built with container-based technology is endowed with many features. In addition, overhauling the architecture of container-technology-driven e-government systems, we observe that securing an e-government system demands quantifying security issues (vulnerabilities, threats, attacks, and risks) and the related countermeasures. Notably, we find that the `Attack Tree` and `Attack-Defense Tree` methods are state-of-the-art approaches in these aspects. Consequently, in this paper, we work on quantifying the security attributes, measures, and metrics of an e-government system using Attack Trees and Attack–Defense Trees—in this context, we build a working prototype of an e-government system aligned with the United Kingdom (UK) government portal, which is in line with our research scope. In particular, we propose a novel measure to quantify the `probability of attack success` using a risk matrix and normal distribution. The probabilistic analysis distinguishes the attack and defense levels more intuitively in e-government systems. Moreover, it infers the importance of enhancing security in e-government systems. In particular, the analysis shows that an e-government system is fairly unsafe with a 99% probability of being subject to attacks, and even with a defense mechanism, the probability of attack lies around 97%, which directs us to pay close attention to e-government security. In sum, our implications can serve as a benchmark for evaluation for governments to determine the next steps in consolidating e-government system security.

**Keywords:** e-government; security; quantification; container; Docker; Kubernetes; Attack Tree; Attack–Defense Tree; risk matrix

## 1. Introduction

An electronic government or e-government is a crucial part of a smart city that makes use of ICT to change the relationship between citizens and governing bodies, enterprises, and other government departments, while focusing on improved government service qualities, closer interactions, and more effective governmental operations. This helps to improve the quality of public services and to enhance the security and privacy of data, applications, and services [1,2].

We find that almost all the cities in the world currently support online services through e-government portals. In particular, e-government portals provide online services to citizens via different websites and mobile applications using modern technologies, such as the internet, cloud computing on computers, and mobile devices (for example, Amsterdam

Smart City (https://amsterdamsmartcity.com/ (accessed on 16 January 2023)) [3,4], the UK government portal (https://www.gov.uk/ (accessed on 16 January 2023)) [5,6], and Singapore government Citizen Connect portal (https://www.citizenconnectcentre.gov.sg/ (accessed on 16 January 2023))).

Particularly, the goal of a smart city and an e-government system [1,7–10] is to ensure better quality of life for the stakeholders. As we observe, there is a wide-spread adoption of e-government services; however, e-government systems and their services encounter myriad challenges. Specifically, the challenges include accessibility, efficient responses, reliability, availability, security, privacy, usability, maintainability, and reusability. They also suffer from a low penetration of ICT, particularly in developing areas [1,2]. Therefore, a good ICT infrastructure is important to have for effective and efficient facilitation of government information flow.

On the other hand, the technical challenges encountered by e-governments include a high availability of systems and services; the storing and processing of mammoth amounts of data; the security of systems, applications and services; the privacy of data and services; the scalability of applications and services; the load-balancing of applications and services; high performance expectations; and disaster recovery. Therefore, we need to rise to the aforementioned challenges of e-government and address the associated issues.

In this paper, we conduct our analysis on the enhancement of security of e-government systems, applications, and services. We dive deeply into the underlying technologies of e-government to understand the architectures and principles to achieve our goal. We observe that the traditional computing architectures, strategies, and solutions are impotent to fulfill the prevailing requirements and demands of smart governance.

Hence, the emerging technologies, such as containers (Docker [11–16], rkt [17,18], and many others), container-orchestration tools (Docker Swarm [19–22], Kubernetes (also known as K8S) [12,20,23], Nomad [20], and many others), and serverless computing [24,25] (OpenFaaS [25,26], Kubeless [25,27], OpenWhisk [25,28,29], and many others), undertake these challenges and overcome them by using this advanced approach for computing [8,9,30,31].

We found that the UK government portal was developed with Docker, Kubernetes, and Amazon Web Services (AWS), and the experience is more stable and consistent than its previous version (formerly developed with hypervisor-based technology) [30,31]. In addition, we see that Docker can help build a secure, intelligent, privacy preserving, cost-effective, and efficient e-government system [8,9,30,31]. Moreover, integrating Kubernetes and Serverless with the system helps to stabilize the system [32].

We focus on the theoretical and practical perspectives of container-technology-driven e-government systems. In addition, we deep-dive into identifying the issues attached to different e-government systems and addressing them. As stated earlier, in this paper, we exclusively pay attention to the security aspects. As such, we particularly work on quantifying the security of e-government systems and services. Specifically, security (risk) quantification is desired to be measured and reported—particularly in terms of financial (loss) quantity.

In addition, it is essential to measure and report in terms of the quality of service, service level agreement, privacy, integrity, safety, confidentiality, unreliability, unavailability, downtime, vulnerability, threats, attacks, and more [33–39], which can help enhance the quality of e-government systems and services end-to-end. To specify, among the various aspects of security quantification, we carefully limit our discussion points to the quantification of security attributes, measures, and metrics, i.e., the probability of attack success. In achieving the goal, we explore the following specific areas in great detail:

- Analyzing and evaluating traditional and current e-government systems and services. Understanding their underlying infrastructures and principles, and identifying the challenges and issues attached with them (Section 3).
- Understanding the underlying architecture, principles, and applications of container technologies (e.g., Docker and Kubernetes) toward the development of e-government systems. In addition, empirically analyzing them toward the formation of the in-

frastructure of e-government systems and deployment of applications and services (Section 4).

- Understanding and analyzing the prominent and standard security quantification mechanisms (for quantifying security attributes, measures, and metrics), such as `Attack Tree (ATree)` [40,41] and `Attack-Defense Tree (ADTree)` [42–44]. In addition, carefully comparing and deciding on tools for modeling the Attack Tree and Defense Tree (Section 2). Furthermore, introducing security risks, threats, attacks, and vulnerabilities of Docker, Kubernetes, and serverless computing toward the overall security of e-government systems and services (Section 4.6).
- Proposing a quantitative measure to logically compute the probability of risk (attack) or countermeasures. Confining our analysis to be rational and rigorous in the context (Section 5).
- Performing risk analysis for each layer of the e-government system (specifically, we follow the architecture outlined in Section 4.1) and, thereafter, computing the risk probability of the whole system. Finally, analyzing how the quantification can reflect the end-to-end enhancement of e-government systems and services (Section 6).

The rest of this paper is structured as follows. In Section 2, we show our efforts in analyzing the related work dedicating to Attack Trees and Attack–Defense Trees. Section 3 presents the architectures, principles, and underlying technologies of certain prominent electronic government systems. In addition, we comparatively analyze them regarding the addressing of challenges and issues. Section 4 presents a simplified architecture of a container-technology-driven e-government system. Note that it is based on our study of different container-based e-government systems and smart governance.

Section 4 briefly presents the possible/potential security risks, threats, and vulnerabilities of container technologies toward the security flaw analysis of the container-technology-based e-government systems. To this end, Section 5 demonstrates the proposition of a novel probability quantification method toward security analysis. Section 6 explores the security quantification of each layer in e-government systems. We calculate the risk probability of the entire system and have a discussion about the results. In the last section, our conclusions and future work are presented.

## 2. Related Work

As stated earlier, in this paper, we study the security quantification of e-government using viable mechanisms, and thus we begin with introducing the commonly used approaches for analyzing security issues. We find that the notable approaches are `Attack Trees (ATrees)` [40,41], `Attack-Defense Trees (ADTrees)` [42–44], `Fault Trees` [33,34,45], `Reliability Block Diagrams` [33,34], `Threat Models` (Threat Models https://owasp.org/www-community/Threat_Modeling (accessed on 16 January 2023)) [46], `Risk Trees` (RiskTrees https://risktree.2t-security.co.uk/) (accessed on 16 January 2023) [47], and `Bayesian Networks` [48–51], among others.

We observe that the aforementioned approaches perform similar kind of analyses dealing with security issues, threats, vulnerabilities, exploits, attacks, and risks. Notably, an `Attack Tree` is used for modeling the possible attack scenarios or threats in a system [40,41], and an `Attack-Defense Tree` [42–44] helps us to model the defense of those attacks or threats. On the other hand, a `Fault Tree` helps us analyze the root causes of a failure while computing the unavailability, downtime, and so on [33,34,45].

In contrast, a `Reliability Block Diagram` is specialized for analyzing component reliability, which may lead to the success or failure of a system [33,34]. Similarly, a `Threat Model` works on analyzing security threats and/or the absence of associated defensive measures [46]. Likewise, a `Risk Tree`, built on the Attack-Tree method can help us monitor security risks [47]. Similarly, a `Bayesian Network` can help us perform fault analysis, safety analysis, reliability modeling, risk analysis, risk prediction, and more [48–51].

Among the methodologies illustrated above, we choose the Attack Tree and Attack–Defense Tree modeling techniques, since these are quite in line with our scope and well-

established compared with others. Furthermore, as stated earlier, we perform quantitative and qualitative analyses concerned with security, such as Attack Tree and Attack–Defense Tree, are prominent for measuring the `probability of attack` and `minimum cost of attack`. Notably, in this context, our goal is to quantify the `probability of attack` success as stated earlier.

To this end, in this section, we perform a literature review to perceive the principles of Attack Tree and Attack–Defense Tree . In addition, we analyze the popular and commonly used tools and frameworks for modeling Attack Tree and Attack–Defense Tree. Moreover, we analyze how the tools are used by the community in security quantification. Finally, based on the analysis, we perform our security quantification.

### 2.1. Principles of Attack Tree and Attack–Defense Tree

Mauw et al. [52], 2005 offered a formalization of the concepts with a grounding on attack suites. The formalization helps us to learn the principles of Attack Tree and Attack–Defense Tree. In addition, we see how to use this method to model risks so that we can build a tree correctly. In addition, we learn a more formal interpretation of Attack Trees.

Similarly, Terrance R Ingoldsby [53] analyzed threats using an Attack Tree. The author introduced a large set of samples to build attack scenarios and precisely showed the quantitative analysis with a table and line chart. In addition, the author discussed the countermeasures and controls. From the work, we can obtain a comprehensive understanding of the Attack Tree, including the origins, concepts, structures, analysis, and calculations. Notably, we can learn how to model threats in the context of a certain situation. It immensely helps us utilize the Attack Tree and Attack–Defense Tree methods in constructing attack scenarios for an entire system or an application.

#### 2.1.1. Attack Tree

In 1999, Schneier came up with the Attack Tree (ATree) [40,41] to represent and evaluate potential security risks on systems. He used graphical, mathematical, and structured decision tree symbols to model possible attacks and systematically classified the ways a system can be attacked. The Attack Tree is constructed from the opponent's perspective. Note that we play the role of an attacker in creating a lucid Attack Tree.

This is a multi-level tree composed of roots and leaves as shown in Figure 1 (notably, the last node `Countermeasure 1` is for defending against corresponding attacks—particularly those not included in the Attack Tree). In an Attack Tree, the overall goal of the attack is marked as the `root` node on the top of the tree. In contrast, `leaf` nodes are sub-targets that an attacker may implement in order to execute an attack. Usually, when we first create a model, we do not need to pay attention to how to protect the system.

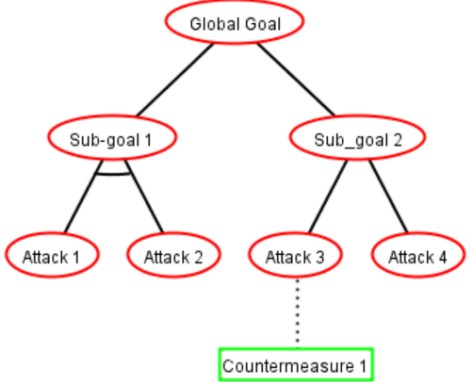

**Figure 1.** Attack–Defense Tree structure.

2.1.2. Attack–Defense Tree

The basic formalism of an Attack Tree does not consider the defense mechanism. Therefore, the Attack–Defense Tree (ADTree) uses defense measures (also called countermeasures) to expand the Attack Tree. This generates a graphical mathematical model of multi-stage attacks with related countermeasures [42]. Note that a defense node is placed in the Attack–Defense Tree, and an Attack–Defense Tree helps us study the effects of defense mechanisms using measures, such as the attack/defense cost and attack probability. [43,44].

As such, each node belongs to an attacker (represented by a red ellipse) or a defender (represented by a green square) as shown in Figure 1. A countermeasure is an action to prevent an opponent from achieving the goal. This concept is used to explain the relationship between an attacker who attempts to attack the system and a defender who protects the system [42].

*2.2. Tools for Modeling Attack Tree and Attack–Defense Tree*

At the beginning of our research, we thoroughly investigated and carefully compared some existing Attack Tree modeling tools. Then, we divided them into two categories:

- **Commercial**: One is commercial applications that require a fee, such as `AttackTree+` from **Isograph**, `SecurlTree` from **Amenaza Technologies**, and `RiskTree` from **2T Security**.
- **Open Source**: The other is open source applications (for example, `ADTool` [54,55], `Ent` [56], and `SeaMonster` [57]).

From these two sets of tools, we did not use the commercial tools since we would need to pay for them. In our research, we decided to pick the best one from the open source tools, and finally we selected the ADTool.

In Table 1, we show a comparison of the three aforementioned open-source tools so that readers can more easily comprehend the differences among them. Principally, ADTool is the most in line with our expectations and requirements.

**Table 1.** Attack Tree modeling tool comparison (open source).

| Names / Features | Ent | SeaMonster | ADTool |
|---|---|---|---|
| Last update time | May 2016 | November 2016 | November 2017 |
| Operating environment | Only for Mac system | No system limitations | No system limitations |
| Functions of the tool | 1. Create tree with text inputs<br>2. Saving and loading | 1. Create tree by dragging graphics in Palette<br>2. Saving and loading<br>3. Can set countermeasures | 1. Create tree with text inputs<br>2. Saving and loading<br>3. Can set countermeasures<br>4. Create tree by mouse right clicks<br>5. Perform quantitative analysis |

*2.3. Applications of Attack Tree and Attack–Defense Tree*

Using the MyProxy system, a remote service that stores user proxy credentials, Saini et al. built an Attack Tree to illustrate potential attacks to the system [58]. MyProxy is an important security subsystem of Globus, a distributed computing paradigm. With their practical, high-level guidance, we can understand how to attack a remote service system and learn how to construct an Attack Tree in the context of cloud computing. Since our study involves cloud technology, their work is very timely to help us build an Attack Tree for breaching the security of distributed computing systems.

Any internet application or system needs to ensure secured access, particularly for binding government elections, which need to count every vote. To determine the precise probability of a threat toward an internet voting system, Pardue et al. [59], built a threat tree for the system. In their paper, they presented that there are three high-level threats for voting systems: voting equipment attacks, voting process attacks, and insider threats.

According to these three types of threats, they modeled them as independent branches in a threat tree and outlined the related threat sub-actions for each type of threat in a numbered list format. The authors defined hierarchical subordination by indentation and outline numbering. Most importantly, they used the OR and AND operations to indicate whether the relationship between each threat was optional or not. Although their analysis focused on a threat tree instead of an Attack Tree, it gives us inspiration for performing an access attack on a system. From their work, we can also learn how to divide risks into different classifications. This helps us to perform a similar analysis.

After analyzing several related works of Attack Trees in general applications or systems, we found that risk assessment quantification is also important for ICT systems with cloud computing. Consequently, we observe that an Attack Tree can play a significant role in this process. In this context, S Tanimoto et al. [60,61] performed an analysis. They showed that, although cloud computing has been studied thoroughly in recent years, more investigations have been focused on the services side rather than on the security side.

In their first study [60], they extracted the risk factors of the cloud based on the risk breakdown structure (RBS) method [62,63] and proposed a group of measures. Based on the results of a security risk investigation on cloud computing conducted by Jon Brodkin [64], they conducted a risk analysis [60]. Their analysis helps us become aware of the risks of cloud computing and comprehensively guides us in understanding cloud security from a social viewpoint. However, in their first work, they did not provide a quantitative evaluation of the risks.

To fill the gap, S Tanimoto et al. conducted further study [61] and proposed a risk quantification matrix. In their extended study, they calculated a risk value for each risk factor and its countermeasures with their risk formula. With the risk matrix method, they simplified the quantification. The analysis helps us learn how to calculate a relatively scientific and reasonable risk value for a risk in a quantification analysis. However, they did not use an Attack–Defense Tree method to visualize the risks and measures.

We suggest that, in a sense, an Attack–Defense Tree can more intuitively help us understand the ranks and relationships between various risks. In addition, it is essential to have a fine granularity of the risk matrix. To extend this work, in our quantification, we improve the risk matrix model from a 4-division to a 9-division cell.

Overall, we analyze a significant number of works in the context of modeling threats, attacks, and vulnerabilities. The analysis helps us perform the security quantification of e-government systems in an elegant manner.

## 3. Analysis of E-Government Systems

In this paper, we focus on analyzing the architecture of e-government systems of different countries and cities, including Canada, the UK, the USA, China, Korea, and Amsterdam [3–6,65,66]. This helps us understand the architecture of real-world e-government systems and helps us identify the challenges or issues associated with them. In addition, we perform a literature review to realize the architecture of different e-government systems. Consequently, we analyze the associated challenges and issues as well as how the systems address them.

We find that the traditional e-government systems are attached to a myriad of challenges and issues, including single point of failure, poor availability, absence of disaster recovery, inefficient deployment and scaling, expensive implementation, poor security and privacy, and low performance. Now, we present two prominent e-government architectures and explore how they help address the aforementioned challenges and issues. The architectures are:

- Container-technology-based e-government systems [8,30,31,67–72].
- Hypervisor-technology-based e-government systems [2,73,74].

### 3.1. Container-Technology-Based E-Government Systems

Our analysis over the real-world e-government systems shows that the UK government portal was developed with container-based technology. Their tech blog states that Docker is more stable and consistent than hypervisor-based technology [30,31]. We performed a literature review to analyze the Docker-driven e-government systems and found that it has better user experience, efficiency, cost-effectiveness, safety, privacy, security, and more [8,67–72]. Most importantly, we found that container-orchestration frameworks, such as Docker swarm, Kubernetes, and Mesos, are often used to monitor and manage the underlying IT infrastructure and applications in smart cities and e-governments.

### 3.2. Hypervisor-Technology-Based E-Government Systems

We observe that some of the e-government systems have been built on top of hypervisor-based technology, such as OpenStack (https://www.openstack.org/ (accessed on 16 January 2023)), a cloud management platform [75]. However, they are gradually migrating to container-based platforms [8,30,31,67–72]. The OpenStack-based e-government architecture is related to some of the aforementioned challenges. We know that hypervisor-based virtualization has its drawbacks and limitations; therefore, OpenStack suffers the same. To analyze this, we built an OpenStack cluster and found that it was tedious to deploy a simple application. Most importantly, addressing the formerly stated challenges is complex, not very efficient, and not optimal.

Our preliminary analysis shows that an e-government system can be designed and built in a better way when adopting container-based virtualization. Therefore, we focus on leveraging containerization and serverless computing to devise an e-government system. Thereafter, we work on addressing the challenges and issues of the e-government systems. All in all, we focus on building a secure, intelligent, privacy-preserving, cost-effective, efficient e-government system using container-based technology since it provides better support and ensures better quality of service.

## 4. Container-Technology-Driven E-Government Systems

In this section, we briefly demonstrate the architecture of container-technology-driven e-government systems, deployment of a Kubernetes cluster (which helps us understand the Kubernetes architecture freely), building a working prototype of an e-government system, security quantification aspects, and security state and analysis in e-government.

### 4.1. Architecture of Container-Technology-Driven E-Government Systems

Our literature review on container-technology-driven e-governance finds that the end-to-end architecture follows the bottom-up approach and can be divided into multiple distinct layers [8,30,31,67–72,72] as shown in Figure 2.

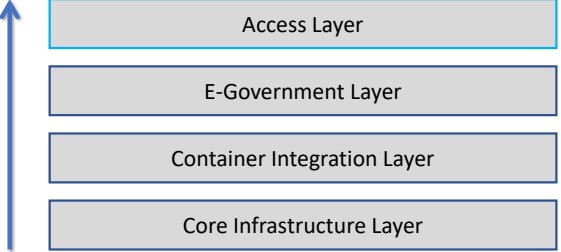

**Figure 2.** Architecture of a container-technology-driven e-government system.

- Core infrastructure layer: This is the first layer that manages and provides various infrastructural components. The component of this layer is an e-government data center.

- Container-integration layer: This is the second layer built on top-of the core infrastructure layer. In particular, the Docker container (Docker, rkt, and many others) is integrated with the infrastructural systems, and the Docker cluster is built. Usually, container-orchestration platforms, such as Docker swarm and Kubernetes, are used to form a cluster. Further, serverless computing frameworks are also deployed in the cluster to deliver services with minimal management effort.
- E-government layer: This is the third layer built on top of the cluster. This layer is in charge of providing e-government services through various government portals and websites. Docker images are built for the services and deployed following the features of Docker. In addition, different data-persistent features are used for data storage and processing. In addition, different databases can be integrated with the Docker image for data processing.
- Access layer: This is the top layer of architecture of the e-government system. Users can access the government services seamlessly with all platform support provided by this layer. Since the containerized applications are light-weight, they can be readily accessed from a thin client.

We found that, when using container technology, we can better address the challenges and issues associated with availability, disaster recovery, deployment, scaling, rolling update, expensive implementation, security, privacy, performance, and others. In addition, we can manage and monitor the e-government systems and services efficiently. To evaluate, validate, and consolidate our analysis, we built a Kubernetes cluster as discussed in the following section and consequently performed the analysis.

As we know, Kubernetes is an open-source system by Google. It performs automated deployment, scaling and managing containerized applications within and across computer clusters. It is a production-grade container-orchestration platform that can easily manage containers [12,23,76]. Kubernetes is the most popular and dominant container-orchestration platform in the community [32]. In addition, we observe that Kubernetes has been used for e-governance [67,68,70–72]. In this paper, we use the Kubernetes container-orchestration framework for our analysis.

### 4.2. Building a Kubernetes Cluster with Kubeadm

Kubeadm is the official default installation method, which is for high-availability cluster installation [77]. We observe that a cluster with Kubeadm can be used in a production environment. For our analysis, we deployed a single master and two worker nodes in a Kubernetes cluster using Kubeadm. Note that we can add as many nodes as we want (if more nodes are required). We installed Kubernetes on top of Ubuntu VMs.

We built the Kubernetes cluster to empirically analyze the issues and the underlying infrastructure of e-government systems and services. In the following section, we present the Kubernetes architecture, which is the container-integration layer on top of the core infrastructure of the e-government system. We deployed one of the most popular serverless computing frameworks, OpenFaaS, with the Kubernetes cluster.

### 4.3. Kubernetes Architecture

After deploying the Kubernetes, we found a cluster. The cluster is composed of multiple nodes and can be divided into two groups—`master nodes` and `worker nodes`. Figure 3 is a holistic structure of Kubernetes.

The master node is the control unit of the cluster; it works with worker nodes, including the sub modules, such as APIServer, Scheduler, Controller Manager, ETCD, and Kubectl.Worker nodes are used to deploy applications in containers called pods. The components of a worker node include Kubelet, Kube-proxy, Pod, ReplicaSet, Deployment, and Secret.

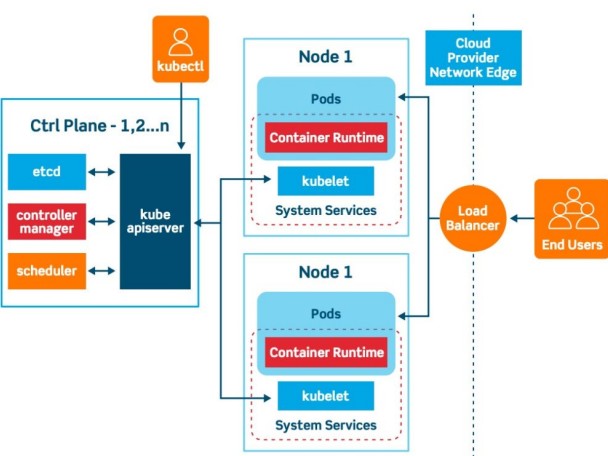

**Figure 3.** Kubernetes cluster architecture [78].

There are many other important components in Kubernetes that are useful for application development, deployment, and maintenance. We refer to official documentation of Kubernetes (https://kubernetes.io/docs/home/ (accessed on 16 January 2023)) for the details.

### 4.4. Working Prototype of an E-Government System

As stated earlier, we analyze a set of current and prominent of e-government systems to perceive their underlying structures and technologies. In sum, we observe that UK government portal developed with Docker, Kubernetes, and Amazon Web Services (AWS) is in line with our research scope. Notably, our Kubernetes cluster integrated with serverless OpenFaaS is aligned with the UK Government portal, so we take it as our working e-government prototype and perform security analysis on it.

### 4.5. Security Quantification Aspects

We found that, without adequate security measures, there can be huge revenue losses. In addition, quality of service, service level agreements, privacy, integrity, security, confidentiality, reliability, availability, downtime, vulnerabilities, threats, attacks, and many other aspects are reduced [33–35,37,38,45]. Therefore, security (risk) quantification is important. In particular, it is to be measured and reported in terms of:

- Financial (loss) quantity.
- Service level agreement, quality of service, privacy, confidentiality, integrity, safety, reliability, availability, unavailability, downtime, vulnerability, threat, attack, and many more.

To this end, security quantification is the defining, structuring, and quantifying security toward measuring the effectiveness of security-related deployments. In particular, it is the process of identifying security vulnerabilities, attacks, threats, and risks; assessing them; and then validating them using mathematical modeling techniques to measure and analyze the available security data. this is a mechanism to accurately represent the security environment that can be used to infer informed security investments and viable measures for risk mitigation [33–35,37,38,45,53,79]. Therefore, in our analysis, we focus on the factors mentioned herewith. Notably, as stated earlier, in this paper, we work to quantify the `probability of attack` success as demonstrated in the following sections.

### 4.6. Security in E-Government

E-government security is a prime concern in numerous aspects, including data, service, and functions. Security flaws are associated with data leakage, denial-of-service attacks, vulnerabilities in legacy functions, risks in over-privileged functions, and more. Therefore,

security analysis is crucial in e-government. To preserve the security of the system (e-government applications, services, and associated data), we should first analyze the security vulnerabilities, threats, attacks, and risks of underlying technologies, such as Docker and Kubernetes.

From the security structure of Docker and Kubernetes, we found that they provide fine-grained security and strong physical isolation of containers. In addition, our study finds that Docker containers are fairly secure and that they can defend themselves from different external security attacks and internal security vulnerabilities, such as kernel exploitation, poisoned images, compromised secrets, denial of service (DoS) attacks, man-in-the-middle (MITM), ARP spoofing, leaky system calls, sharing roots, and file systems that are not isolated [80–82].

However, we need to perform a thorough analysis, since new security threats and vulnerabilities are reported every year. Note that the security vulnerabilities with a vulnerability score can be traced from the repository of CVE Details (The Ultimate Security Vulnerability Datasource https://www.cvedetails.com/index.php (accessed on 16 January 2023)). A vulnerability score helps us compute the risk matrix for computing the probability of attack or countermeasures as proposed in Section 5.1. The computation of the risk matrix is shown in Section 6 while analyzing the `probability of attack` with an Attack Tree and Attack–Defense Tree, specifically following the architecture, deployment, and principles outlined in this section.

## 5. Measures for Security Analysis

As stated earlier, our research aim is to quantify security for the container-technology-driven e-government systems, and thus our top priority is to obtain a measure to model and quantify the risks or attacks. We need to extract or find potential risks or attacks through vulnerabilities, and then we show them vividly and make them visible for our readers. Finally, we chose to use the Attack–Defense Tree method because of its unique advantages and ADTool for modeling as stated earlier.

Now, we need to consider how to quantify the risks or attacks when we construct the tree. Specifically, we need calculate a risk value of a selected attribute domain, such as the `probability of success` and `minimal cost for the proponent`. Therefore, a key question is how to assign an appropriate value to a risk. Here, we propose a novel measure to calculate a risk value in the `probability of success` attribute domain, in which the value represents the probability of a risk as well as a successful attack. As far as we know, we are the first to propose this quantitative measure and we elaborate it in the following.

### 5.1. Proposed Quantifying Measure

In our security quantitative analysis, we advance a new quantifying measure combining `risk matrix`, `standard normal distribution`, and `probability density function`, to assess possibility of risks reasonably and scientifically. Our measure focuses on estimating possibilities, so among various attribute domains of Attack–Defense Tree method, we narrow down our security quantification analysis to the `probability of success` domain. The proposed quantification measure mainly consists of three steps:

1. Calculate the matrix value with respect to evaluation factors.
2. Normalize the matrix values into the standard normal distribution.
3. Find the probability according to the probability density function.

#### 5.1.1. Risk Matrix

A risk matrix [83] is designed to analyze and quantify certain factors, such as risk, in enterprise and it is used as a general method [84]. Referring to the means of risk matrix formulation in [61], we attempt to utilize the method in our analysis. Essentially, a risk matrix helps us visualize a depiction of risks and we can use it to assess risk levels. Here, to improve the granularity, we adopt a 9 division model ($3 \times 3$) as a risk matrix for the quantification. Since we have to quantify both attacks and countermeasures, we create two

kinds of risk matrices with four different evaluation factors [38]. For risks, we create our matrix in keeping with the degree of danger and the frequency of occurrence shown in Table 2. In Table 3, for countermeasures, the matrix is built based on the cost level and the degree of effectiveness, which show the level of a defense method.

**Table 2.** Risk matrix of attacks.

| Matrix Value | | Occurrence Frequency | | |
| --- | --- | --- | --- | --- |
| | | Low (1) | Mid (2) | High (3) |
| Danger Degree | Low (1) | | | |
| | Mid (2) | | | |
| | High (3) | | | |

**Table 3.** Risk matrix with countermeasures.

| Matrix Value | | Effectiveness Degree | | |
| --- | --- | --- | --- | --- |
| | | Low (1) | Mid (2) | High (3) |
| Cost Level | Low (1) | | | |
| | Mid (2) | | | |
| | High (3) | | | |

Next, the evaluation factors are divided into three levels including High (3), Medium/ Mid (2), and Low (1). We assign the values to them based on our knowledge and understanding. It is worth pointing out that there is no need to assign a fixed value (3, 2, 1) to every risk or countermeasure. Instead, the values of the evaluation factors can be assigned in a continuous domain from 1 to 3. Thus, it is apparent to understand that the domain of the final evaluation result in matrix is from 1 to 9 and we call this result the `matrix value`. The matrix value of a attack or countermeasure is calculated by different formulas, which are determined by the corresponding evaluation factors.

Accordingly, the matrix value for attack expressed in Equation (1):

$$\text{Value} = \text{Danger Degree} \times \text{Occurrence Frequency} \tag{1}$$

Similarly, the matrix value for defense expressed in Equation (2):

$$\text{Value} = \text{Cost Level} \times \text{Effectiveness Degree} \tag{2}$$

In our quantification measure, after getting the matrix value, to deduce the approximate value as a probability, we need to perform the following steps.

### 5.1.2. Normal Distribution

In probability theory, a normal distribution, also called Gaussian distribution [85], in a variate $X$ with mean $\mu$ and variance $\sigma^2$ is a continuous probability distribution with identically distributed random variables. The parameter $\sigma$ is the standard deviation and when the $\mu = 0$ and $\sigma = 1$, it is the simplest form called the standard normal distribution. The shape of normal curve takes the mean as the symmetry axis shaped like a bell and both ends of the curve gradually and evenly drop to the left and right sides but never intersect the horizontal axis. The area between the curve and the horizontal axis is always equal to 1. We can use a function to represent this curve. Then, it means the integral of this function from negative infinity to positive infinity is 1 (shown in Figure 4).

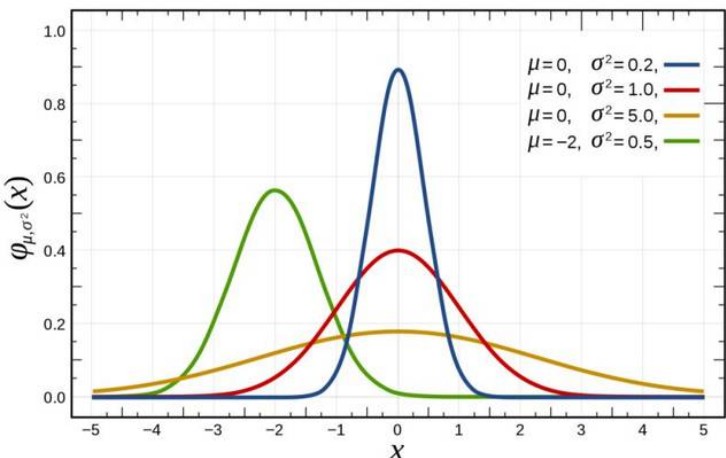

**Figure 4.** Normal distribution [85,86].

　　From the risk matrix method, we can calculate the degree of a risk or countermeasure. Specifically, by multiplying evaluation factors, we find the matrix value. Note that, for countermeasures, the value indicates the level of the defense, which means that the higher the value, the greater the defense. For example, there are some high-level defense measures whose defense performance is fantastic and can almost eliminate most risks; however, the cost is also extremely expensive.

　　This defensive measure will not be adopted by most companies because it is not cost-effective and vice versa. If a countermeasure has extremely low effectiveness, even if it is quite cheap, there are still few companies that will adopt it. It is the same for a risk. In addition, limiting the value on integer, we can exhaust all situations shown in Table 4. We can easily find that the two extreme values (Minimal and Maximal) have the least number of occurrences.

**Table 4.** The matrix of defense with all situations (for an integer).

| Matrix Value | | Effectiveness Degree | | |
|---|---|---|---|---|
| | | Low (1) | Mid (2) | High (3) |
| | Low (1) | 1 | 2 | 3 |
| Cost Level | Mid (2) | 2 | 4 | 6 |
| | High (3) | 3 | 6 | 9 |

　　Combining our knowledge in practice and real world, it is reasonable for us to assume that the distribution of the matrix values, which are continuous random numbers in the domain 1 to 9, conforms to the characteristics of a normal distribution with the mean $\mu = 5$ in the case of a large number of data samples.

### 5.1.3. Probability Density Function

　　After illustrating that the distribution of matrix values is a normal distribution, now we begin to explain how to calculate the probability of each feasible matrix value. We need to use the probability density function to find the possibility. In mathematics, the probability density function [87] of a continuous random variable is a function that describes the probability of the output value of a random variable near a certain value point. In short, this function can calculate the probability of a certain value in a continuous random variable domain. A function meeting the following conditions can be regarded as a probability density function.

$$f(x) >= 0$$

We can easily obtain the probability density function of normal distribution from mathematics papers as shown in Equation (3).

$$P(x) = \frac{1}{\sigma\sqrt{2\pi}} e^{\frac{-(x-\mu)^2}{2\sigma^2}} \tag{3}$$

In order to facilitate the probability calculation for standard normal distribution ($\mu = 0$, $\sigma = 1$), the statistician developed a statistical table named standard normal distribution table. From it, we can quickly find the probability of any certain value that we desire.

Thus, we are required to normalize the normal distribution of our matrix values to a standard normal distribution and there are some steps we need to take for standardization. If a random variable $X$ obeys a normal distribution with a mathematical expectation (mean) of $\mu$ and a variance of $\sigma^2$, it would be recorded as $X \sim N(\mu, \sigma^2)$. If we attempt to calculate the probability of variable $X$ in a traditional method, we need to integrate the function shown in Equation (3), then we can obtain the probability $P(x)$ even though it is a taxing process.

However, if this distribution is standardized, the probability is much easier to calculate. First, we perform a calculation as follows in Equation (4). As the retrieval of this table is only applicable to data accurate to two decimal places, we will only keep two decimal places in calculations. In addition, we list the formulas to calculate mean ($\mu$) in Equation (5) and standard deviation ($\sigma$) in Equation (6) as shown.

$$Y = \frac{X - \mu}{\sigma} \tag{4}$$

$$\mu = \frac{1}{N} \sum_{i=1}^{N} X_i \tag{5}$$

$$\sigma = \sqrt{\frac{\sum_{i=1}^{N}(X_i - \mu)^2}{N}} \tag{6}$$

After this, we convert the variable $X$ that conforms to the general normal distribution into the variable $Y$ in the standard normal distribution recorded as $Y \sim N(0,1)$. Then, we only need to look in the standard normal distribution table, according to its absolute value of $Y$, to get the corresponding probability.

In the following security quantification analysis, we apply the Attack–Defense Tree method to model the attack scenario and perform the quantification with our proposed measure. Therefore, we can analyze the probability of attacks and countermeasures in a more rigorous and precise way.

## 6. Security Quantification of E-Government Systems

In this section, we thoroughly demonstrate the security quantification of e-government systems. Notably, we begin with demonstrating the architecture of e-government systems and then we dive into security quantification as following.

### 6.1. Architecture of E-Government Systems

Based on our literature review, an architecture of a container-technology-driven e-government system is presented in Section 4. Referring to the architecture, we observe that the architecture is divided into multiples distinct layers [8,30,31,67–72,72]:

- Core Infrastructure layer.
- Container Integration layer.
- E-government layer.
- Access layer.

Now, in the following subsections, we show the security quantification of each layer.

### 6.2. Security Quantification of Core Infrastructure Layer

Cloud Computing is a state-of-the-art technology to improve the ability of scaling, flexible, high availability computing infrastructures [88]. Thanks to its advantages, prominent ICT systems have been developed to introduce cloud computing as a technology to alter their infrastructures, such as electronic government system. For example, the Ministry of Internal Affairs and Communications in Japan has a smart cloud study progressing. [60]. In addition, as early as 2000, Maria and Roland proposed that distributed knowledge management is the development trend of electronic government management [89]. Most importantly, we observe that cloud computing helps address the challenges and issues of e-government [90–94]

Therefore, cloud computing, as a technology of distributed information management in e-government system's core infrastructure layer, its security is of paramount importance. In this section, we perform our efforts to quantify the e-government cloud computing system with the Attack–Defense Tree method.

Attack Scenario One

As stated earlier, cloud-based technology is used to build the infrastructure layer of the e-government, so we construct the attack scenario of the layer concerning cloud computing. Expecting to make the research more complete and rigorous, the main subjects of security analysis toward cloud computing are divided into two categories: cloud service provider and the user of the e-government system. Then, we perform a security quantification analysis including modeling and quantifying as follows.

Attack Tree Modeling and Quantification

**Attack Tree Modeling:** We built an Attack Tree to model the potential threats from the perspective of government with cloud computing—the whole procedure is shown in Figure 5. The Attack Tree diagram clearly indicates three directions of our analysis. Systemic threats mainly focus on the risks that may be provided by the cloud service provider, such as Amazon EC2 and Alibaba Cloud. The second analysis direction is the actions threats of users who work in the government and the last one is some risks from outside of these two.

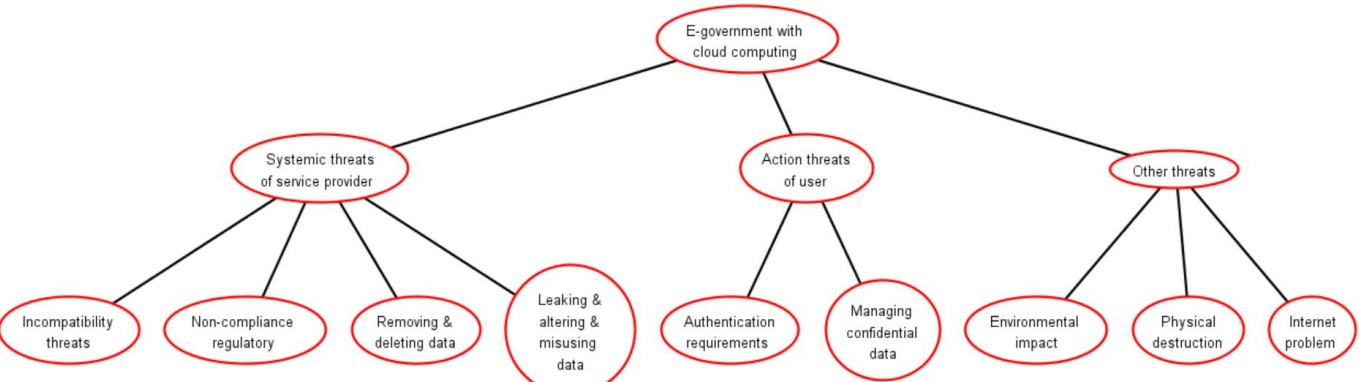

**Figure 5.** Attack Tree for e-government infrastructure with cloud computing.

When the government utilizes cloud computing, there may be an incompatibility risk, such as operating system incompatibility. For example, the platforms of cloud service providers are usually incompatible with certain applications, which makes the migration of data between various platforms difficult. In addition, each cloud system uses different protocols, and some service providers have their own unique specifications, which leads to low compatibility. Moreover, for some business interests, some cloud service providers will have some problems with regulatory non-compliance, which means that they are beyond the scope of their rights.

Another concern is the risk of data security including both unexpected and malicious operations. As for unexpected operations, it is likely to occur the situation of removing data or data deletion after using cloud service. Theoretically, the data saved in the cloud is safe and is replicated on different machines. Hence, the possibility of data loss is low, consequently and most likely, we may not consider to backup at local machine. That means if the cloud occurs accidental faults, users relying on the cloud have some risks.

Since the cloud is an open platform, it may be subject to malicious operations from insiders and outsiders. For insiders, both service providers and users from the government may leak, alter and wrongly use the data. In particular, some of users have strong impact on security who are at the administrator level. In contrast, there also some threats from outsiders, such as competitor's attack and human-made physical damage to the facility.

Now, it comes to the threats caused by user's actions—data in the cloud usually exists in a shared environment, so owners of data should be in charge of who has the right to use the data and which part of data they can use after they obtain access. Access management is one of the most serious issues in cloud computing security [95]. However, a user without specialist knowledge may not proficient in the detailed operation management method, so sometimes, a user may ignore the importance of authentication requirement. Accidentally, strangers may access to some mission critical resources.

Similarly, managing confidential information also needs to be taken into account. It is required to restrict access to only approved users, and protect sensitive data, even if unauthorized personnel coincidentally acquire the possession of some confidential information, it needs to ensure that the data cannot be read immediately. In addition, there is a risk of environmental impact, such as data center destruction by a disaster. In addition, it may also happen that the device is damaged through physical violence or stolen.

In cloud computing, communication is performed via the internet, and this is the backbone of the cloud environment [96]. Thus, internet problems are equally serious, because if we cannot connect to the internet, we cannot perform cloud computing, which means that we cannot access anything, not even our own files. Unfortunately, internet connections are inherently unreliable areas. Apart from that an attacker may perform `deauthentication attack` to refrain users from accessing a cloud [97].

**Quantification with an Attack Tree:** Now, we attempt to quantify the security risks from the standpoint of probability, which indicates the likelihood of a threat occurring. We chose the `probability of success` attribute domain. It is apparently evident that the probability of success is congruent with the probability of a risk occurring because, in this context, regarding the risk, the success implies the presence of the risk instead of avoiding it.

According to our quantitative evaluation method, we chose the risk matrix of attacks. We assigned the values for those two evaluation factors (occurrence frequency and danger degree) first and perform the multiplication to obtain the matrix value. Next, we needed to normalize these matrix values into the standard normal distribution; thus, we calculated the mean and the standard deviation of them. Based on Equation (4), we found the standard value ($Y$) of each matrix value ($X$). Then, we derived the probability with respect to the absolute value of $Y$. Note that we show all the data in Table 5.

- Mean:

$$\mu = \frac{30.53}{9} = 3.39$$

- Standard Deviation:

$$\sigma = \sqrt{\frac{13.11}{9}} = \sqrt{1.46} = 1.21$$

**Table 5.** Risk matrix value for the Attack Tree of a cloud.

| Factors / Risks | Occurrence Frequency | Danger Degree | Matrix Value | Standard Value | Probability |
|---|---|---|---|---|---|
| Incompatibility threat | 2 | 1 | 2 | −1.15 | 0.13 |
| Non-compliance regulatory | 2.6 | 1 | 2.6 | −0.65 | 0.26 |
| Remove and delete data | 1.2 | 2.5 | 3 | −0.32 | 0.37 |
| Misuse data | 3 | 1.7 | 5.1 | 1.41 | 0.92 |
| Authentication requirement | 1.6 | 2.7 | 4.32 | 0.77 | 0.78 |
| Confidential data | 2 | 2.5 | 5 | 1.33 | 0.91 |
| Environment impact | 1.5 | 2.4 | 3.6 | 0.17 | 0.57 |
| Physical destruction | 1 | 1.4 | 1.4 | −1.64 | 0.05 |
| Internet problem | 2.7 | 1.3 | 3.51 | 0.10 | 0.54 |

After finding the probabilities, we can assign these values into the Attack Tree leaf nodes, and then it will automatically calculate the whole probability of successful attack for the e-government system built on cloud computing. The Attack Tree is displayed in Figure 6. Attacking the infrastructure of the e-government system (e-government infrastructure with cloud computing), we can see that the probability of success is 99%.

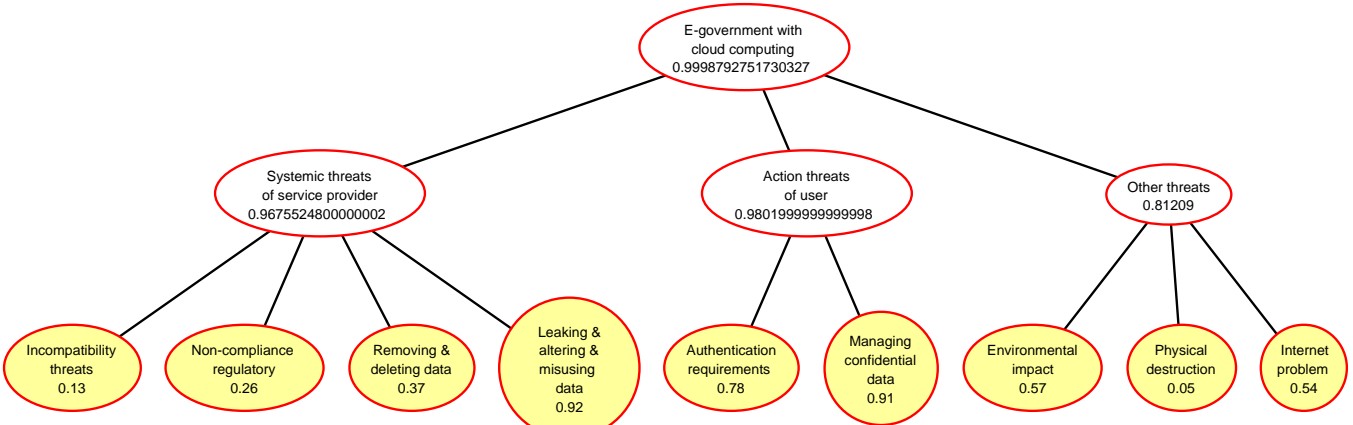

**Figure 6.** Probabilities in an Attack Tree for e-government infrastructure with cloud computing.

Attack–Defense Tree Modeling and Quantification

**Attack–Defense Tree Modeling:** Using an Attack Tree, we indicate the possible security risks in the e-government infrastructure. We need to find attack-related measures to avoid or mitigate them. Consequently, we model an Attack–Defense Tree as shown in Figure 7.

First, if there is an incompatibility threat, it can be resolved easily through the adjustment of users. For example, they can simply change the application so that it cannot cooperate with cloud computing or choose a cloud service provider with high compatibility. In short, the user has the right to choose or adjust. When the government decides to sign a contract with the cloud service provider, the specifications of service should be clearly explained in detail in advance so that the power of the service provider can be constrained and inspected.

If the service provider has some violations, such as non-compliance regulatory, the insurance and compensation are prepared after contracting and obtaining the guarantee of the service provider. Users can back up the data in time or perform a distributed storage of data to reduce losses of removing or deleting data accidentally. In addition, third party surveillance is an effective countermeasure for leaking or misusing data on purpose.

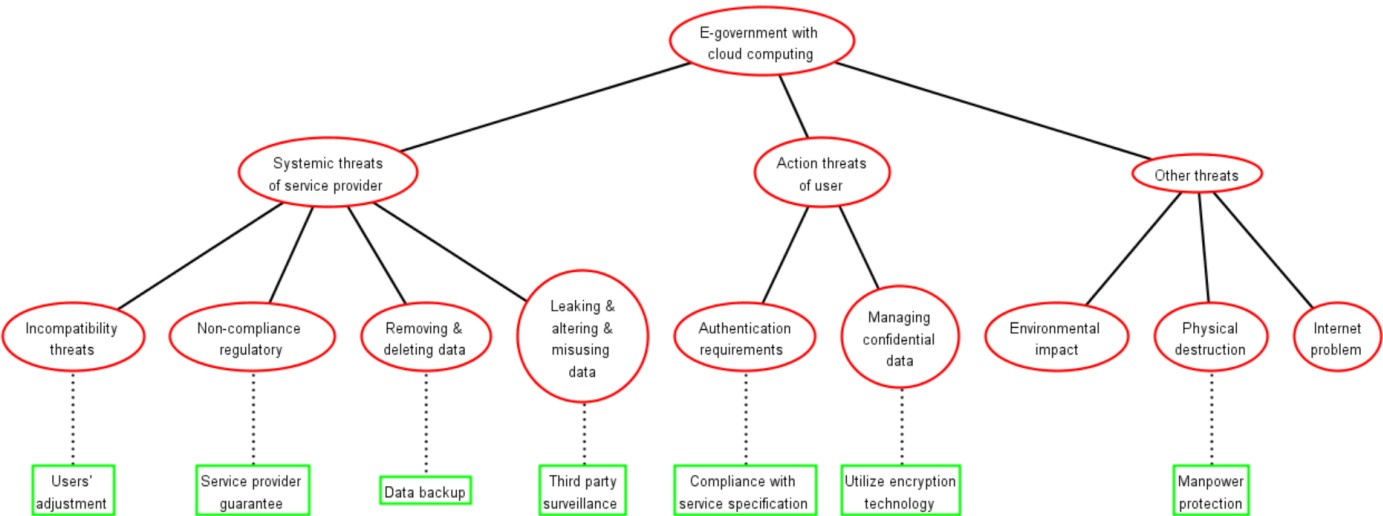

**Figure 7.** Attack–Defense Tree for e-government infrastructure with cloud computing.

It is more safe and convenient to monitor data movement through a third party rather than staff within the government. For insiders, the users can formulate a set of strict rules and regulations to deter those who attempt to endanger security. Furthermore, the users are demanded to comply with service specification, which specifies the correct operations in the cloud, such as the authentication requirements.

Equally, the government can make a cloud instruction manual or let staff learn some corresponding courses to ensure that users can operate correctly in the cloud and have more methods to manage confidential data. However, in the context of security, it is difficult to predict or control the impact of the environment and internet. In fact, human-made physical destruction is also difficult to deal with; however, we can attempt our best to prevent and protect through manpower.

**Quantification with an Attack–Defense Tree:** Here, we take the countermeasures into consideration and make an analysis using the proposed quantification measure, within the same attribute domain as the analysis of the Attack Tree. After assigning the evaluation values (effectiveness degree and cost level) based on our knowledge, we determine the $\mu$ and $\sigma$ of matrix values with the following computation. From Table 6, it is easier for us to know the whole calculation process.

- Mean:

$$\mu = \frac{31.47}{7} = 4.50$$

- Standard Deviation:

$$\sigma = \sqrt{\frac{23.11}{7}} = \sqrt{3.30} = 1.82$$

In this way, as shown in Figure 8, we objectively display the risk reduction after giving the proposed countermeasures. It is worth noting that, although the probability after adding our countermeasures is still somewhat high, which is around 95%, this does not mean that our analysis is wrong. First, we can see that, in the Attack Tree, the probabilities of environmental impact and internet problems as obtained by the proposed measure are both around fifty percent.

This is a general probability, which is the same as the probability of many things in real life, and it is even somewhat low in the context of risk. These two probabilities are reasonable and in line with expectations. However, since these two risks are both difficult to predict and they are extremely uncontrollable, it is too difficult for us to provide efficient countermeasures for them. As such, we do not set defensive measures for these two in the Attack–Defense Tree—we do not consider these two risks.

**Table 6.** Defense matrix value for the Attack–Defense Tree of a cloud.

| Defenses / Factors | Effectiveness Degree | Cost Level | Matrix Value | Standard Value | Probability |
|---|---|---|---|---|---|
| Useradjustment | 3 | 1.2 | 3.6 | −0.49 | 0.31 |
| Providerguarantee | 2.7 | 1 | 2.7 | −0.99 | 0.16 |
| Databackup | 2.8 | 1.5 | 4.2 | −0.16 | 0.44 |
| Third partysurveillance | 2.9 | 2.7 | 7.83 | 1.83 | 0.97 |
| Obeyspecification | 2.3 | 2 | 4.6 | 0.05 | 0.52 |
| Utilizeencryption | 2.6 | 2.4 | 6.24 | 0.96 | 0.83 |
| Manpowerprotect | 1 | 2.3 | 2.3 | −1.21 | 0.11 |

Therefore, we assign zero to them in calculation and regard the probability under ideal conditions as the final probability of this attack scenario. The eventual result is about 72% as shown in Figure 9. Therefore, in ideal conditions, ruling out the influence of the environment and network connection, we found that the probability of a successful attack to the e-government system utilizing cloud computing technology was around 72%.

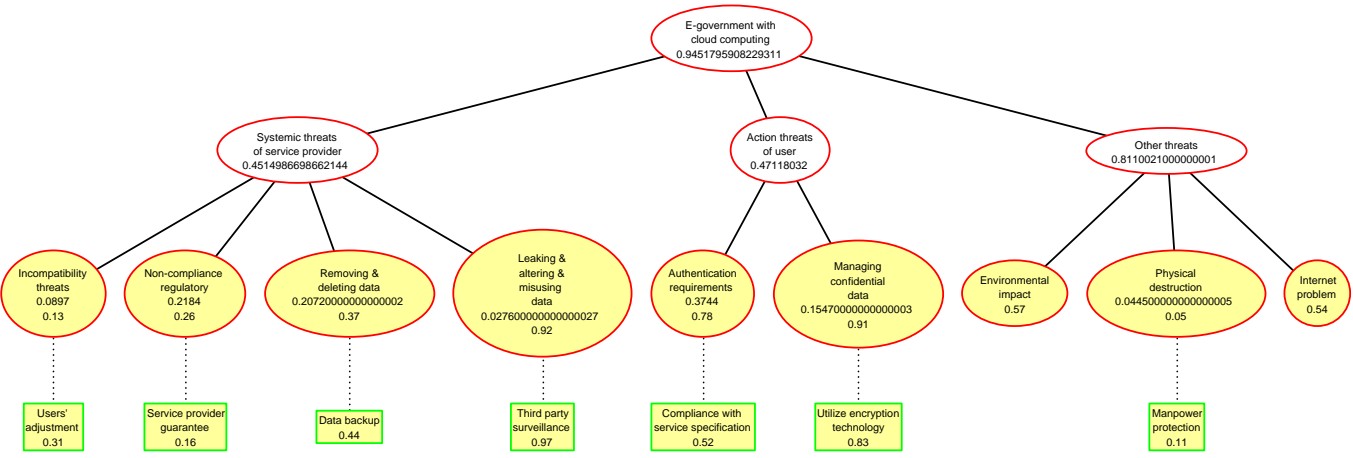

**Figure 8.** Probabilities in an Attack–Defense Tree for e-government infrastructure with cloud computing.

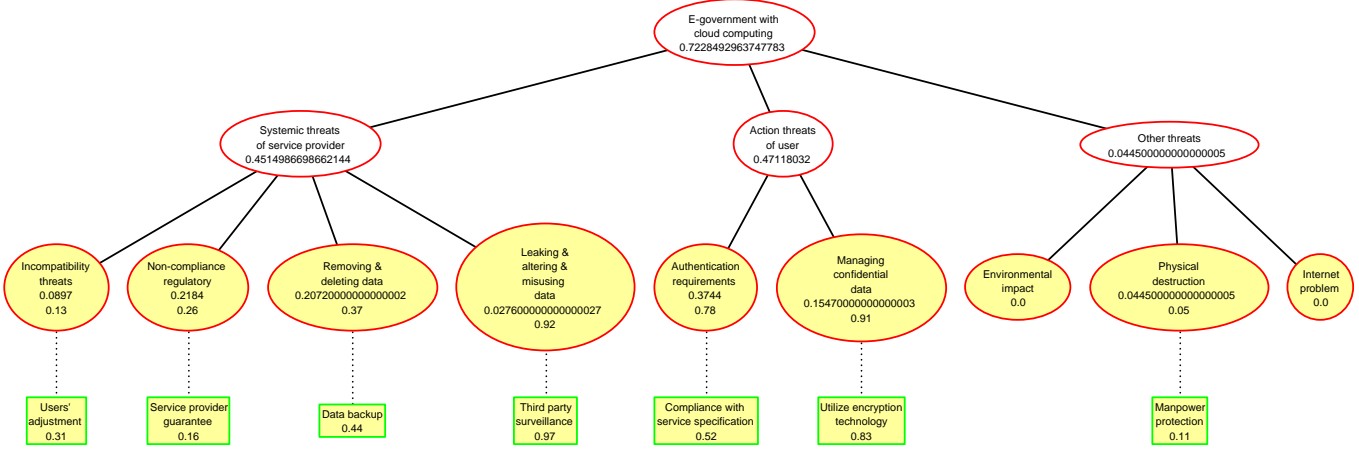

**Figure 9.** Ideal probabilities in the ADTree for e-government introducing cloud computing.

*6.3. Security Quantification of Container Integration Layer*

As stated earlier, the core infrastructure is built on top of Kubernetes. Therefore, we performed analysis of the Kubernetes cluster and its components.

### 6.3.1. Attack Tree of Kubernetes Cluster

Kubernetes is a fast-growing project and is the most popular container-orchestration framework [32]. Many developers move their services forward to Kubernetes because of the flexibility and scalability of the container. However, Kubernetes brings us some security challenges. Therefore, it is important to learn that there will be many security issues in the existing containerized environment, particularly in Kubernetes. To this end, Attack–Defense Trees can greatly help us model the process.

To illustrate the methodology, we constructed a simple Attack Tree based on the MITRE ATT&CK framework [98] and modeled the procedure of attacking the Kubernetes cluster. The MITRE ATT&CK framework is a network attack-related framework covering recognized tactics and technologies.

### 6.3.2. Attack Scenario Two

The subject of this example is to attack the Kube-API server, which marshals all the communication on master node, instead of the entire cluster because of its simplicity to readers and because the less complex tree structure can fit on a single page better. The whole process is shown in Figure 10. According to the structure of the Kubernetes cluster, we find that there are three different attack paths.

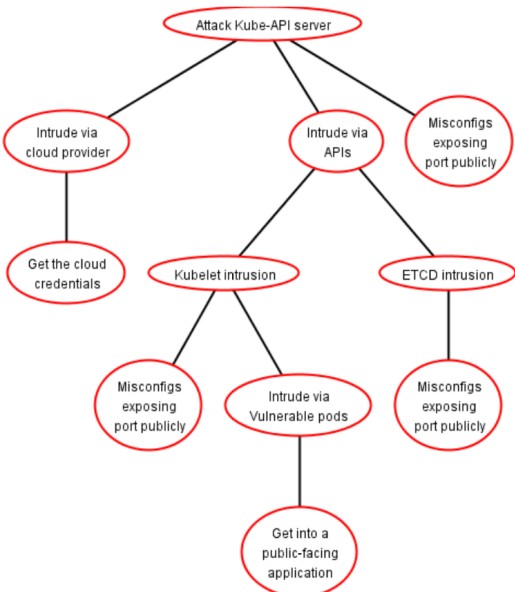

**Figure 10.** Attack Tree for attacking the KubeAPI server.

Attack Tree Modeling and Quantification

**Attack Tree Modeling:** First, attackers can intrude on the cluster's management layer if the Kubernetes cluster runs in a public cloud provider with compromised cloud credentials (for example, EKS in AWS, GKE in GCP, and AKS in Azure.). In addition, we know that the Kube-API server keeps interactive communication with ETCD and Kubelet through the APIs. As a result, attackers can also attack the Kube-API server by invading these two components.

In a case where the worker node has any vulnerable pods that contain containers with public-facing applications, the remote code execution (RCE) vulnerability may be exploited by the attacker. Note that this may result in sending requests to the Kube-API server using the service account credentials if the `service account` (the service account in

Kubernetes provides identity for processes running in a pod; https://cloud.google.com/kubernetes-engine/docs/how-to/kubernetes-service-accounts (accessed on 16 January 2023)) is mounted to the container (the default behavior in Kubernetes).

Another common risk for Kubernetes users is the misconfiguration issue, which makes a `service` (service in Kubernetes is a way of exposing an app running in a pod. https://kubernetes.io/docs/concepts/services-networking/service/ (accessed on 16 January 2023)) that should not be exposed or publicly accessible. For instance, Kubelet allows unauthenticated access to an API, which is exposed by default on port 10250/TCP—this makes it vulnerable and able to be attacked by outsiders. Similarly, insecure configurations of Kube-API server and ETCD, which expose the unexpected ports, lead to external attacks.

**Quantification with an Attack Tree:** We chose the `probability of success` attribute domain to quantify the Attack Tree. In this attribute domain, we have to assign the probability of risks (namely, attacks) for each leaf node in the Attack Tree as stated in the former section in Attack Tree analysis for the core infrastructure layer. Similarly, we assign the values depending on our own understanding and experience. The matrix values, the mean and standard deviation, are computed as shown.

- Mean:

$$\mu = \frac{17.40}{3} = 5.80$$

- Standard Deviation:

$$\sigma = \sqrt{\frac{11.72}{3}} = \sqrt{3.91} = 1.98$$

We present all the data in Table 7, which clearly shows the matrix value ($X$) with its standard value ($Y$) and also displays the probabilities. It is easy to determine that the risk of misconfiguration leading to exposing the port to the public is a high-level risk whose probability of success is 90%.

**Table 7.** Risk matrix value for the ATree of the Kube-API server.

| Factors \ Risks | Occurrence Frequency | Danger Degree | Matrix Value | Standard Value | Probability |
|---|---|---|---|---|---|
| Find cloudcredentials | 1.2 | 3 | 3.6 | −1.11 | 0.13 |
| Exposingport publicly | 3 | 2.8 | 8.4 | 1.31 | 0.90 |
| Public-facingapplication | 2.1 | 2.6 | 5.46 | −0.17 | 0.43 |

From the Attack Tree domain analysis, we found the probability of successfully attacking the Kube-API server. The probability was 99%, and the tree diagram is shown in Figure 11. This means that, if the Kube-API server does not have any protection, it is extremely vulnerable regarding the Kubernets cluster.

Attack–Defense Tree Modeling and Quantification

**Attack–Defense Tree Modeling:** After modeling the Attack Tree for attacking the Kube-API server and performing quantitative analysis, we need to construct an Attack–Defense Tree with the related countermeasures, which is shown in Figure 12.

If we want to mitigate the risk of compromised cloud credentials, we can achieve this by impersonating service accounts. The idea of impersonation is to use one identity A to act as another identity B but without having access to B's credentials, and this is achieved by granting identity A the ability to obtain an access token for identity B [99]. We only grant this permission to identity A, and it should show an access token when it needs to access any resource.

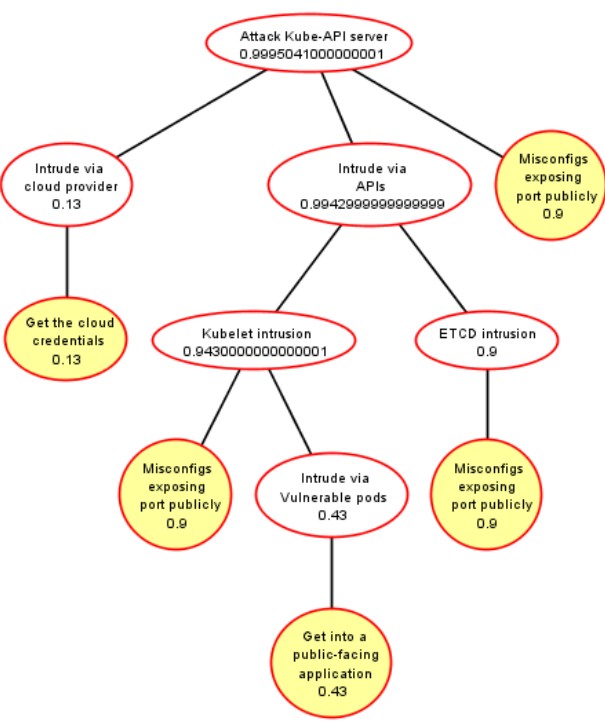

**Figure 11.** Probabilities in the Attack Tree for attacking the KubeAPI server.

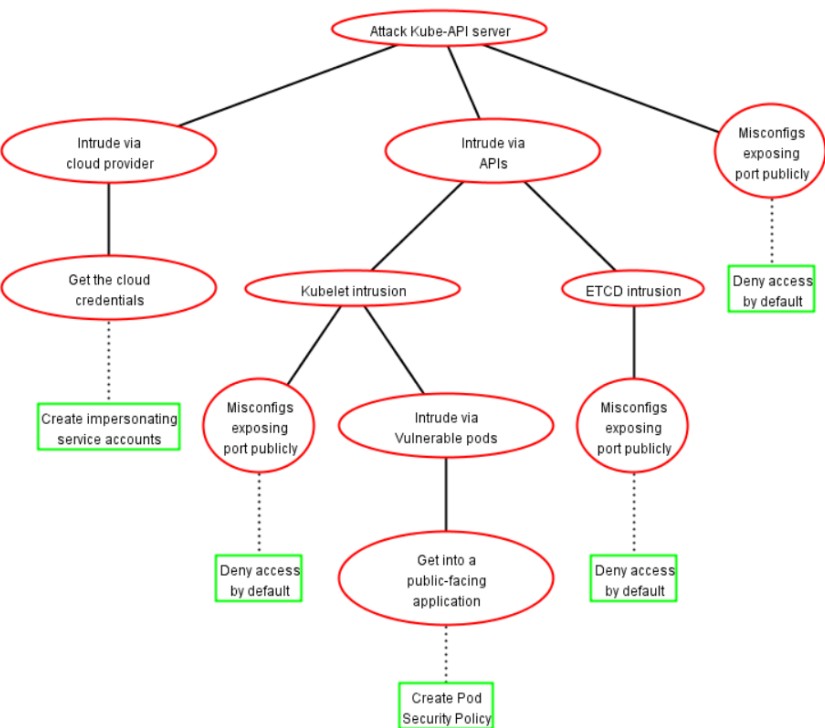

**Figure 12.** Attack–Defense Tree for attacking the KubeAPI server.

As for the misconfiguration issues, the best defensive practice is to deny access by default and allow traffic only explicitly. Moreover, cloud administrators need to check the guidance from the cloud provider and conduct a regular review thereafter to ensure that all services are properly firewalled and not exposed publicly. When it comes to vulnerable pods, we need to ensure that all container images used are the latest and are downloaded from trusted sources. In addition, we can use container-specific automated scanning technologies to scan images for applications, and we also can build pod security policy

rules, which define some security conditions. Thus, if a pod wants to be accepted into the cluster, it must run under conditions.

**Quantification with an Attack–Defense Tree:** In this part, we perform a quantitative analysis of the Attack–Defense Tree with the `probability of success` attribute domain and all the above processes as a complete procedure of security modeling through ADTree. In a similar manner to before, we can see the standard value of each defensive measure in Table 8, and creating an impersonating service account has the highest probability.

- Mean:

$$\mu = \frac{18.32}{3} = 6.11$$

- Standard Deviation:

$$\sigma = \sqrt{\frac{0.8373}{3}} = \sqrt{0.28} = 0.53$$

**Table 8.** Defense matrix value for an ADTree of the Kube-API server.

| Defenses | Effectiveness Degree | Cost Level | Matrix Value | Standard Value | Probability |
|---|---|---|---|---|---|
| Create impersonating service accounts | 2.7 | 2 | 5.4 | −1.34 | 0.09 |
| Deny access by default | 2.9 | 2.3 | 6.67 | 1.06 | 0.86 |
| Create pod security policy | 2.5 | 2.5 | 6.25 | 0.26 | 0.60 |

The ADTree diagram is displayed in Figure 13. There is a great reduction of risk after adding the defensive measures. The automatically computed value in the root node changed to 55%, which was 99% in the Attack Tree. Thus, it reflects that, although we cannot totally eliminate the risk, we can mitigate the impacts through certain effective measures.

*6.4. Security Quantification of the E-Government Layer*

Finishing the analysis of the container-integration layer, which concentrates on Kubernetes, we come to attack the e-government layer in which the public obtains the service via portals and websites. Internet technology holds great potential to improve the interface between public organizations and citizens [100]. With the help of the internet, the relationship between the government, enterprises, and residents has become more coordinated than ever before.

Through the e-government system portal, citizens can access and gain civic information and public services easily, and this also provides a channel for them to communicate with the government so that the citizens can better participate in the management of the government. However, in addition to the benefits, the internet also has many drawbacks. Websites have been continuously targeted by malicious users to acquire monetary gain, and there has been an increase in internet crimes over the years [101]. Although web applications are the cornerstone of the realization of various functions of the e-government system, this is not without threats.

Thus, we perform a security analysis and attempt to determine the risks or vulnerabilities of these portals and websites. We found some knowledge of website vulnerabilities from The Open Web Application Security Project (OWASP) (https://owasp.org (accessed on 16 January 2023)). OWASP is an online community working to develop software and documentation for securing web applications and web services [102,103]. In 2017, OWASP released a list of the most critical website vulnerabilities. This indicates a common agreement about what the most critical web application security risks are [102–105].

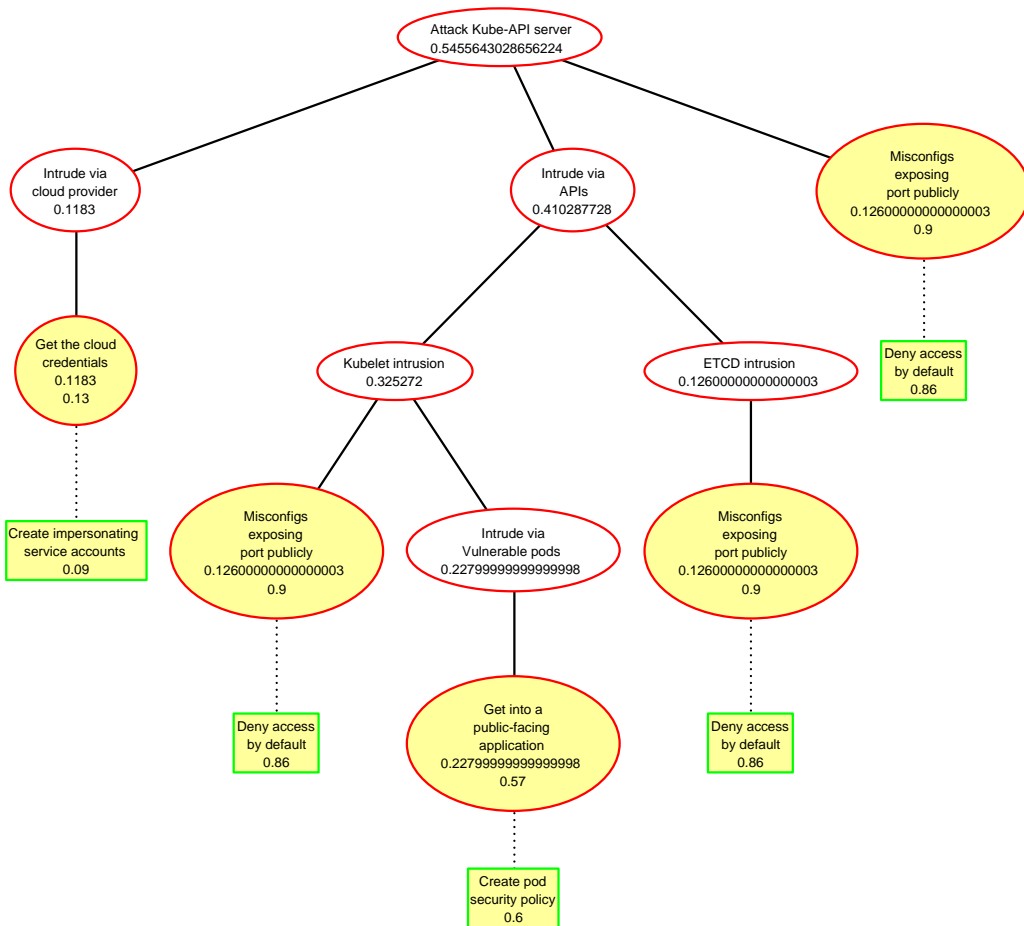

**Figure 13.** Probabilities in the Attack–Defense Tree for attacking the KubeAPI server.

Attack Scenario Three

In this case, the subject of our scenario is the portals and websites of the e-government system, which is the entrance to the public. In accordance with OWASP, we construct our attack scenario, and then we conduct a security quantification evaluating the probability of these vulnerabilities with our proposed quantification measure.

Attack Tree Modeling and Quantification

**Attack Tree Modeling:** The steps are the same as in the above examples. We first present an Attack Tree in Figure 14 to visualize the risks or attacks that have been executed through these vulnerabilities.

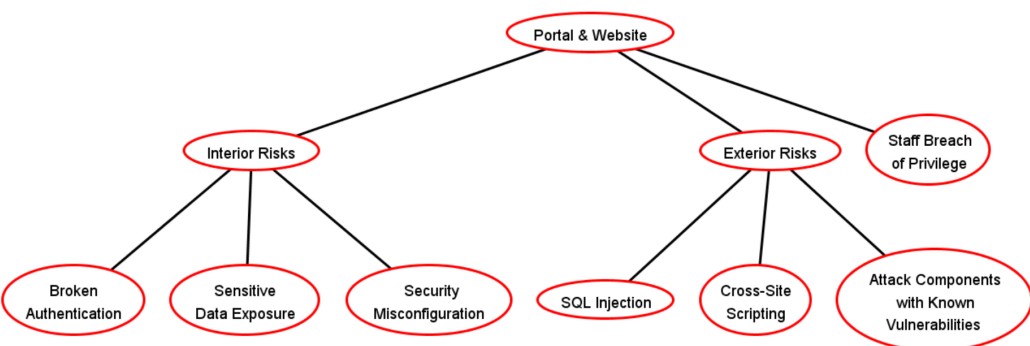

**Figure 14.** Attack Tree for e-government portals.

This is mainly divided into three possible risk types, which consist of interior risks, exterior risks, and the staff breach of privilege. We discuss staff violation of privileges first. This is a common but dangerous risk in many enterprises and organizations because staff have access to the organization's systems, networks, and data, and they can use legally acquired access rights to negatively affect the confidentiality, integrity, and availability of information in the organization's information system. Thus, for internal staff, attacking the website is very simple.

Exterior risks represent methods that may be used by external attackers to attack the website rather than self-existing flaws. SQL injection attacks are very common attacks. These are widely used to attack the database behind web applications. This attack constructs a malicious SQL query or creates a statement exploiting the logic vulnerabilities of the input parameters in target web application, and then the SQL interpreter behind the target web application executes the malicious SQL query and returns confidential data without authorization.

Cross-site scripting (XSS) is a web security vulnerability that allows attackers to execute scripts in the pages provided to other users. In this way, the attackers can steal the cookies stored on the client or other sensitive data used by other websites to identify the client.

Attackers can even impersonate legitimate users to interact with the website. Some components included in the application run with full permissions unexpectedly (for example, third libraries, development frameworks, and other modules in the application). Consequently, it is easy to attack a website if its applications and APIs that exploit components have known vulnerabilities.

On the contrary, interior risks refer to defects that exist in the websites or portals themselves, and these vulnerabilities are exploited by malicious people. In many websites, the functions of authentication and session management are weak, and attackers can assume using manual combinations or can use the help of automated tools with a password list [106–108] (Note that we can use the Crunch (Crunch https://tools.kali.org/password-attacks/crunch (accessed on 16 January 2023)) tool to generate a password list).

In addition, some websites have weak protection for sensitive data, such as banking information, health information, and user accounts. Thus, it is likely to cause a financial loss when attackers obtain these exposed sensitive data. Moreover, as with the former analysis in Kubernetes, misconfiguration is also the most ordinary risk issue in website security when security settings are defined improperly. For example, insecure default configurations, default account access, and open cloud storage all give attackers an opportunity to attack websites.

**Quantification with an Attack Tree:** We chose the level of each threat from High (3), Medium or Mid (2), and Low (1). For example, most portals have a weak or even broken authentication management; thus, the frequency grade is 3, which is at a high level. In addition, the probabilities corresponding to other risks are also all presented in Table 9. Moreover, we assign these data to our Attack Tree as shown in Figure 15.

- Mean:

$$\mu = \frac{32.40}{7} = 4.63$$

- Standard Deviation:

$$\sigma = \sqrt{\frac{12.59}{7}} = \sqrt{1.80} = 1.34$$

**Table 9.** Risk matrix value for an Attack Tree of portals.

| Factors<br>Risk | Occurrence<br>Frequency | Danger<br>Degree | Matrix<br>Value | Standard<br>Value | Probability |
|---|---|---|---|---|---|
| BrokenAuthentication | 3 | 1 | 3 | −1.22 | 0.11 |
| DataExposure | 2.6 | 2.3 | 5.98 | 1.01 | 0.84 |
| SecurityMisconfiguration | 2.7 | 2.1 | 5.67 | 0.78 | 0.78 |
| SQLInjection | 2 | 3 | 6 | 1.02 | 0.85 |
| Cross-SiteScripting | 2.1 | 2.6 | 5.46 | 0.62 | 0.73 |
| VulnerableComponents | 1.5 | 2.2 | 3.3 | −0.99 | 0.16 |
| Staff Breachof Privilege | 1.3 | 2.3 | 2.99 | −1.22 | 0.11 |

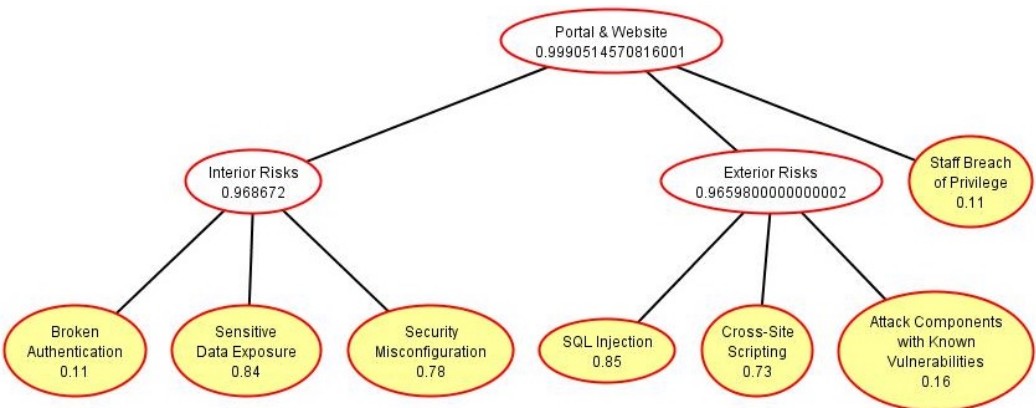

**Figure 15.** Probabilities in the Attack Tree for e-government portals.

Attack–Defense Tree Modeling and Quantification

**Attack–Defense Tree Modeling:** Although we encounter many website-associated risks, we have some viable means to avoid or mitigate their impacts. In real life, a risk has multiple countermeasures; however, we only list one as a representative in our Attack–Defense Tree as shown in Figure 16.

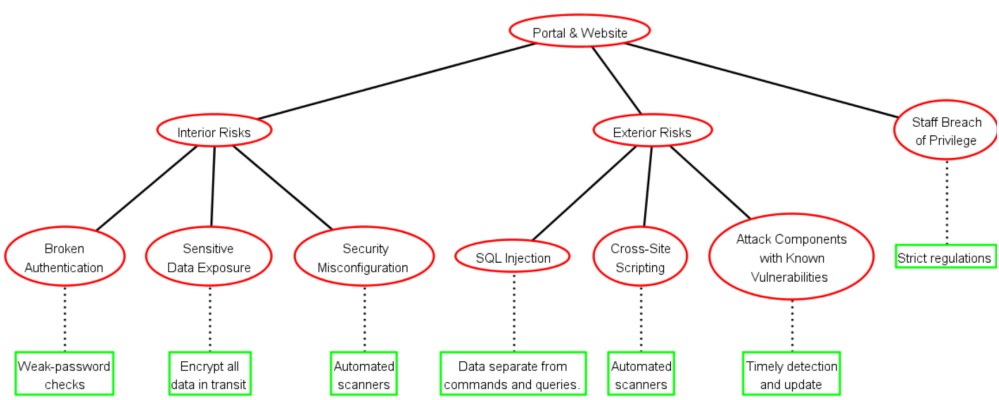

**Figure 16.** Attack–Defense Tree for e-government portals.

When we deal with the issue of staff breaches of privileges, we can only do our best to improve laws and regulations and formulate severe penalties so as to minimize the possibility of such risks. On the other hand, data separation from command and queries should be implemented, when we attempt to mitigate SQL injection attacks, and we also can perform input validation at the server side. To mitigate the cross-site scripting (XSS) risk, we can use context-sensitive encoding to separate the browser content from untrusted data. In addition, we also can apply scanner tools to find the XSS risks automatically.

Timely detection and updates are a good way to prevent the risks caused by components with known vulnerabilities. Furthermore, acquiring components over secure links that are released from official sources is also a good countermeasure to these risks. Weak-password checks are essential to mitigate the broken authentication risk, such as asking for a minimal length and complexity of passwords and limiting failed login attempts. Where possible, conducting all data encryption in transit is a way to prevent sensitive data exposure.

To prevent misconfiguration, the preferred option is to utilize an automated scanner to inspect insecure configurations, such as default passwords. Similar to insecure components, reviewing and updating regularly is always helpful.

**Quantification with an Attack–Defense Tree:** Likewise, we need to quantify the impact of defensive measures so that we can have a clear contrast as before. After computing, we found that $\mu$ is equal to 3.61 and $\sigma$ equals 1.45. In Table 10, we show the standard values obtained from normalization and the probability after searching the statistics table.

- Mean:

$$\mu = \frac{21.66}{6} = 3.61$$

- Standard Deviation:

$$\sigma = \sqrt{\frac{12.58}{6}} = \sqrt{2.10} = 1.45$$

**Table 10.** Defense matrix value for an Attack–Defense Tree of portals.

| Factors<br>Defenses | Effectiveness<br>Degree | Cost<br>Level | Matrix<br>Value | Standard<br>Value | Probability |
|---|---|---|---|---|---|
| Weak-passwordchecks | 2 | 1 | 2 | −1.11 | 0.13 |
| Encrypt alldata in transit | 2.2 | 1.3 | 2.86 | −0.52 | 0.30 |
| Automatedscanners | 2.6 | 2.2 | 5.72 | 1.46 | 0.93 |
| Separate data and queries | 2.7 | 2 | 5.4 | 1.23 | 0.89 |
| Timely detectionand update | 2.3 | 1 | 2.3 | −0.90 | 0.18 |
| Strict regulations | 2.6 | 1.3 | 3.38 | −0.16 | 0.44 |

Assigning the data into the Attack–Defense Tree, we find the success probability of the entire attack scenario. From Figure 17, we can see there is still a 75% success probability to perform an attack on the portals; however, compared to the probability in the Attack Tree (99%), the risk has been clearly reduced.

*6.5. Security Quantification of the Access Layer*

The access layer offers various approaches for stakeholders to obtain government services with quick access. Today, people have access to the internet from their home, office, and even remote areas [109]. There are a myriad of different devices for the public to access the internet, such as digital televisions, desktop personal computers, laptops, tablet computers, and mobile phones.

Since, in the e-government system, the internet is the channel of communication between the government and citizens, essentially, almost any device that can connect to the internet can access government services. In addition, every device has an amount of different potential risks. For example, we may be troubled with network attacks that also include sundry paths.

What is more troublesome is the risk from individuals. For example, a personal device may be stolen, thereby, creating a risk of privacy leakage. In addition, some people do not set up identity authentication for their devices so that anyone can use their electronic accounts on the internet as long as they can find the device. The carelessness of people can greatly affect the calculations of risk probability.

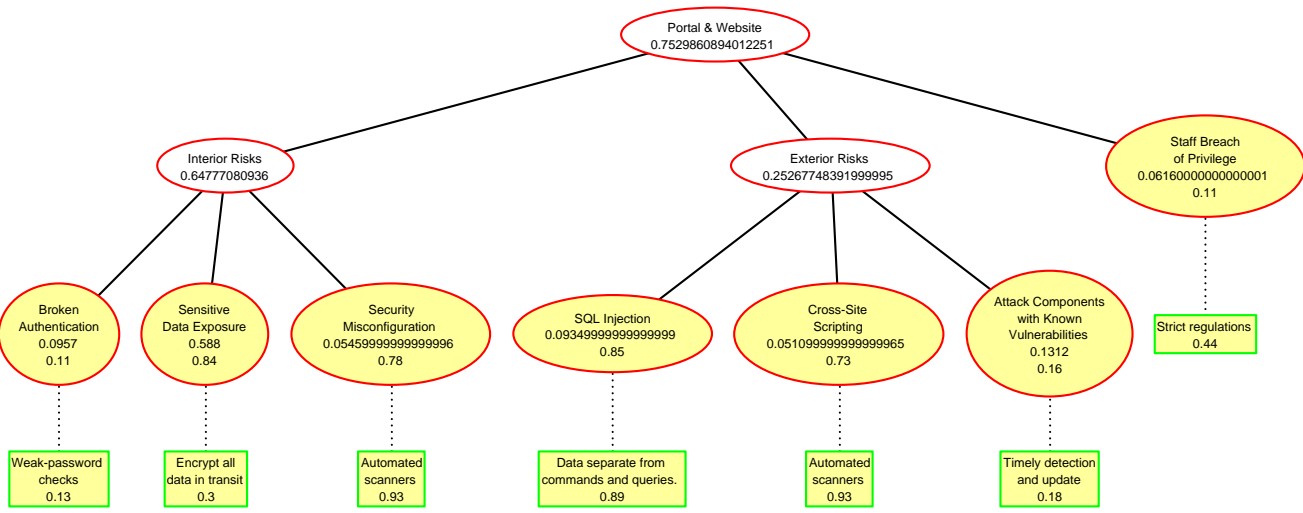

**Figure 17.** Probabilities in the Attack–Defense Tree for e-government portals.

Thus, although it is convenient for users to gain the services of government, it is problematic to construct attack scenarios for this layer because there are many possible ways to obtain access and in particular, there is the uncontrollability of human beings. Moreover, compared to other intangible layers, this layer is more accessible for people to understand as it is related to various physical devices. For these reasons, we did not perform an in-depth study and quantification analysis on this in our paper.

Attack Scenario Four

However, we still generally built an Attack–Defense Tree to describe this process in Figure 18. We generally divided the risk into three types: individual threats, network attacks, and others. Here, we simply list some potential risks, such as non-authenticated devices, and countermeasures, such as using antivirus software to protect from risks.

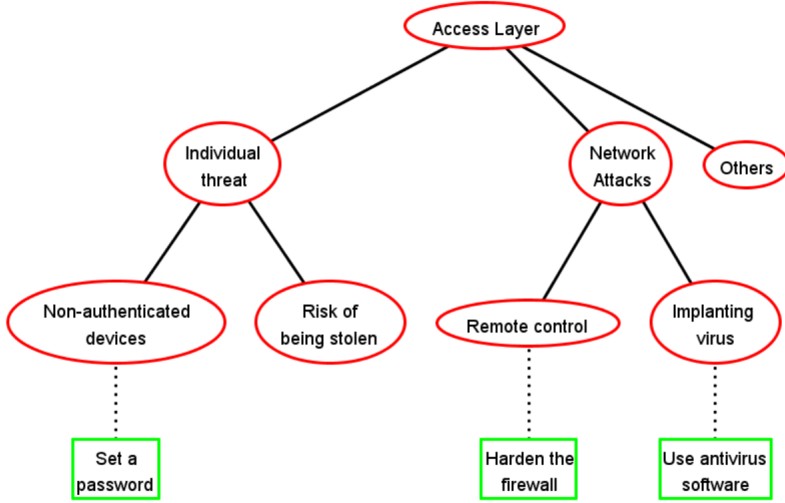

**Figure 18.** ADTree for the access layer.

## 6.6. Toward a More Secure E-Government System

Here, we provide a discussion about the above security quantification results and give some recommendations on creating a more secure e-government system. The results of probability in the aforementioned tables of both Attack Trees and Attack–Defense Trees are summarized in Table 11. The attack scenarios that we established are consistent with the architecture of an e-government system consisting of four layers. However, we mainly

focus on the first three layers to represent the risk probability of the entire e-government system. The reasons for excluding the access layer were stated in detail previously.

From the table, we find a satisfactory reduction of the risk probability when we add the defensive measures. The risk reduction in the container-integration layer is the most significant, and the remaining two layers have around the same degree of decline. The results also indicate that the quantification method that we proposed makes sense.

**Table 11.** Quantification results of each layer.

| Layers<br>Factors | Core<br>Infrastructure Layer | Container<br>Integration Layer | E-Government<br>Portal Layer |
|---|---|---|---|
| Probability in ATree | 99% | 99% | 99% |
| Probability in ADTree | 72% | 55% | 75% |

Now, we attempt to determine the risk probability of the container-technology-driven e-government system after quantifying the risk probability in its three most important layers. As we know now, an attacker has a 72% success rate to attack the core infrastructure layer, which uses cloud computing technology. There is a 55% probability for a malicious person to successfully perform an attack on the container-integration layer, which utilizes the Kubernetes cluster to manage containers, where the applications of the e-government system run. The probability of an attack without obstacles to the portal layer is 75%. In short, as long as one of the three layers is successfully attacked, it is equivalent to the e-government system being successfully attacked. We record the probability of it not being attacked as $F(x)$. Thus, we calculate the risk probability with following equations.

$$F(x) = (1 - 0.72) \times (1 - 0.55) \times (1 - 0.75) = 0.0315 = 3.15\% \tag{7}$$

$$P(x) = 1 - F(x) = 0.9685 = 96.85\% \tag{8}$$

Equation (7) finds the risk-free probability of the e-government system, which is merely 3.15%, and $P(x)$ in Equation (8) represents the risk probability of a container-technology-driven e-government system. According to this result, we can confidently state that the system is still insecure. There is an urgent demand for us to enhance the security aspects of e-government systems. Therefore, we need to find more effective ways to defend against risks or develop a safer system. According to our analysis, we have reasons to strongly suggest that the public pay greater attention to the security issues of e-government systems.

## 7. Conclusions and Future Work

### 7.1. Conclusions

In this paper, we advanced an innovative and distinct quantification measure in our aim to contribute to the security quantification of the container-technology-driven e-government systems. As far as we know, this work is the first to combine the `risk matrix method` and a `normal distribution` in threat analysis, and we further used the `probability density function` to quantify the probability of risk. The analytical results demonstrate that the numerical probability parameters that we obtained from the proposed measure are justifiable and meaningful. We analyzed a set of current use cases of e-government systems—in particular, the UK government portal—which is in line with our research scope.

As such, we built a working prototype of an e-government system and obtained the foundation to build attack scenarios and perform security risk analysis on the e-government system. As stated earlier, we applied the Attack–Defense Tree method to model attack scenarios by constructing ATrees and ADTrees in analyzing the domain of `probability of attack success`.

We calculated the probability of each risk (attack) and countermeasure and then precisely demonstrated the influence of the Attack Tree and Attack–Defense Tree, which

succinctly and vividly indicated the reduction of the risk probability with countermeasures. Finally, we analyzed our security quantification, which infers that it is urgent to pay attention to the enhancement of e-government security.

### 7.2. Future Work

We summarize this paper with a quantified security evaluation with respect to our proposed quantification measure. However, we did not assess and validate the quantification accuracy of this measure. As such, we have a set of plans to do so in our future extension:

- We will focus on strictly evaluating whether our method is strongly reasonable and on justifying its practical value in the security field. For example, we can compare the security quantification results of our method with others to see whether the data can preserve coherence or not for the same security case. In addition, some other attribute domains can be discussed regarding the security quantification.
- We observe that, even though the system is built on top of the most cutting-edge technologies (such as Docker, Kubernetes, Serverless Computing, and other state-of-the-art systems), the underlying technologies have security flaws. Therefore, after analyzing these potential risks more rigorously, we can advance a better system in the future. We observe that not only the security issues but also the underlying technologies have adverse impacts on the performance and dependability, e.g., poor load prediction and autoscaling, cold start, and issues attached with zero downtime, rolling updates, and load balancing. Notably, we are currently working on addressing these issues, and we have had good results in dealing with the issues mentioned herewith.
- Moreover, although we introduced known defensive measures for almost every risk in our analysis, the risk probability is still very high. Thus, we can attempt to determine new and better defensive measures or technologies in this field in the near future. Moreover, we are planning to work on vulnerability quantification using Fault Trees and safety quantification using Reliability Block Diagrams and to thereafter perform a comparative analysis.
- In this paper, the Attack Tree security quantification was limited regarding empirical evaluation and validation. In the future, we can work on performing empirical evaluations and validations of every attack and the related countermeasures. In particular, we need to perform penetration testing to evaluate and validate our findings and analysis. To perform penetration testing, we need to have a clear understanding of the security patterns, root causes, exploits, possible fixes, and more.

  In addition, we need to know about the details of the system configuration for the distinct security patterns, root causes, exploits, and/or possible fixes. Moreover, we need to have in-depth knowledge and experience working with the required tools and techniques for the specific security-related operations. The saving grace is that `Kali Linux` [110] and `Parrot Security OS` [111] are integrated with many security testing tools, which can help us greatly. However, these are not a panacea, and thus we will have to investigate many of the cases for ourselves.

  Notably, we performed system configuration for some of the distinct security cases attached to Docker and Kubernetes (particularly those that were not aligned with `Kali Linux` and `Parrot Security OS`). Consequently, we performed penetration testing for them. Furthermore, we plan to perform penetration testing of the other security issues and threats discussed in this paper. This will help us have a better security analysis and, finally, will help us build an e-government system in a better way. In addition, it will help us detect unidentified security flaws and direct us in devising better countermeasures.

**Author Contributions:** Conceptualization, S.K.M., T.T., S.K., K.K., H.M.D.K. and K.N.; Methodology, S.K.M., T.T. and K.N.; Validation, S.K.M. and T.T.; Formal analysis, S.K.M., T.T., S.K., K.K., H.M.D.K. and K.N.; Investigation, S.K.M. and T.T.; Resources, S.K.M. and T.T.; Writing—original draft, S.K.M. and T.T.; Writing—review & editing, S.K.M., T.T., S.K., K.K., H.M.D.K. and K.N.; Visualization, T.T.; Supervision, S.K.M.; Project administration, S.K.M.; Funding Acquisition, S.K.M. All authors have read and agreed to the published version of the manuscript.

**Funding:** This work was supported in part by the Science and Technology Development Fund of Macao, Macao SAR, China under grant 0033/2022/ITP and in part by the Faculty Research Grant Projects of Macau University of Science and Technology, Macao SAR, China under grant FRG-22-020-FI.

**Data Availability Statement:** Not applicable.

**Acknowledgments:** Authors gratefully acknowledge funding sources. The authors also would like to thank the anonymous reviewers for their quality reviews and suggestions.

**Conflicts of Interest:** The authors declare no conflict of interest.

## Abbreviations

The following abbreviations are used in this manuscript:

| | |
|---|---|
| ICT | Information and Communication Technology |
| VM | Virtual Machine |
| OS | Operating System |
| K8s | Kubernetes |
| ATree | Attack Tree |
| DTree | Defense Tree |
| ADTree | Attack–Defense Tree |
| CNCF | Cloud Native Computing Foundation |
| GKE | Google Kubernetes Engine |
| AKS | Azure Kubernetes Service |
| GCP | Google Cloud Platform |
| EKS | Elastic Kubernetes Service |
| AWS | Amazon Web Services |
| RCE | Remote Code Execution |

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
