# Peer review of "Security Quantification of Container-Technology-Driven E-Government Systems"

_electronics, doi:10.3390/electronics12051238_

Round 1

Reviewer 1 Report

1. The abstract does not clearly state the specific e-government system being analyzed or the context in which the proposed measure will be applied, making it difficult to fully understand the potential impact of the research. Please improve this point in the abstract as well as throughout the manuscript. 

2. The connection between the proposed measure and the consolidation of e-government security is not clearly explained.

3. In the introduction and abstract, it would be good to provide any information on the results of the proposed measure, making it easy to assess the effectiveness of the approach.

4. Please clearly mention any limitations or potential drawbacks of the proposed measure, making it difficult to fully evaluate the approach.

5. It is not clear if the proposed measure has been tested and validated, making it difficult to assess its reliability and generalizability.

6. The paper could benefit from more explanation of the specific e-government system that the authors are analyzing, and how the proposed measure will be applied in a real-world scenario

7. The connection between the proposed measure and the consolidation of e-government security could be explained more clearly.

8. The tree-based approach is often integrated with Bayesian approach. In this regard, a few recent studies have been published. For the purpose of more comprehensive review, please discuss the following studies: 

A modified Bayesian network to handle cyclic loops in root cause diagnosis of process faults in the chemical process industry

Safety analysis in process facilities: Comparison of fault tree and Bayesian network approaches

An integrated risk prediction model for corrosion-induced pipeline incidents using artificial neural network and Bayesian analysis

Dynamic safety analysis of process systems by mapping bow-tie into Bayesian network

Author Response

We would like to sincerely thank the editor and reviewers for providing insightful and constructive feedback. The paper has been revised in the light of reviewers’ comments. Our responses are described in the following, and the changes applied in the revised manuscript are highlighted for the reviewing process’s convenience.

We are uploading (a) our point-by-point response to the comments (below) (response to reviewer), (b) an updated manuscript with red highlighting indicating changes, and (c) a clean updated manuscript without highlights (PDF main document).

We would like to state that we have tried to address all concerns, completely. As suggested, we improve the writing and organization of the paper greatly. We hope that the respected editor and reviewers will find the responses satisfactory. Newly added texts are written as the red text in the improved manuscript. We would like to inform that minor updates are not red marked.

Reviewer 2 Report

The article titled "Security Quantification of Container Technology Driven E-Government Systems" analyzes different e-government systems and highlights that an e-government system built with container-based technology comes with many features. The authors, reviewing the architecture of e-government systems based on container technology, observe that the protection of an e-government system requires the quantification of security problems (vulnerabilities, threats, attacks or risks) and the related countermeasures . In particular, they find the Attack Tree and Attack-Defense Tree methods to be pioneering approaches in these respects. In this paper, the authors work on quantifying security attributes, measures, or metrics using the Attack Tree and the Attack-Defense Tree. Specifically, they propose a new measure to quantify the probability of an attack's success using the risk matrix and normal distribution. Probabilistic analysis more intuitively distinguishes the level of attack and defense in e-government systems. Furthermore, it infers the importance of strengthening security in e-government systems.

The one proposed in the article by the authors can serve as a point of reference and evaluation for the government to determine the further steps in consolidating the security of the e-government system.

Overall it's a good job each section is well articulated. The references and conclusions are good and justify the work proposed in the article.

I suggest a review of the English language

Author Response

(The authors gave the same response as above.)

Reviewer 3 Report

The manuscript presents research related to improving the security of e-governance systems in smart cities. In this aspect, an e-governance system built with container-based technology is proposed, and its advantages are proven by simulating different types of attacks.

In this way, the work is useful and has a great impact on improving the quality of life and, accordingly, on the development of modern society. The manuscript is on a very topical subject and has great potential for development. My main observations and comments are as follows:

- not all figures are of the required quality. I ask the authors to review them carefully and edit them;

- it would be useful, in addition to the advantages in the conclusion section, for the authors to comment on the disadvantages of the proposed electronic management system. In this way, a complete picture of its qualities, capabilities and application limitations will be obtained;

- I recommend commenting on the economic effect of the implementation of the management system, so that the presented solution can be more convincingly accepted by the readers.

Author Response

(The authors gave the same response as above.)
